# Population Structure and Genetic Diversity Analysis in Sugarcane (*Saccharum* spp. hybrids) and Six Related *Saccharum* Species

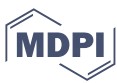

**Haizheng Xiong** [1,†], **Yilin Chen** [1,†], **San-Ji Gao** [2], **Yong-Bao Pan** [3,*] and **Ainong Shi** [1,*]

1   Department of Horticulture, University of Arkansas, Fayetteville, AR 72701, USA; hxx007@uark.edu (H.X.); yc046@uark.edu (Y.C.)
2   National Engineering Research Center for Sugarcane, Fujian Agriculture and Forestry University, Fuzhou 350002, China; gaosanji@fafu.edu.cn
3   Sugarcane Research Unit, USDA-ARS, Houma, LA 70360, USA
\*   Correspondence: yongbao.pan@usda.gov (Y.-B.P.); ashi@uark.edu (A.S.)
†   H.X. and Y.C. are co-first authors.

**Abstract:** Sugarcane (*Saccharum* spp. hybrids) is one of the most important commercial crops for sugar, ethanol, and other byproducts production; therefore, it is of great significance to carry out genetic research. Assessing the genetic population structure and diversity plays a vital role in managing genetic resources and gene mapping. In this study, we assessed the genetic diversity and population structure among 196 *Saccharum* accessions, including 34 *S. officinarum*, 69 *S. spontaneum*, 17 *S. robustum*, 25 *S. barberi*, 13 *S. sinense*, 2 *S. edule*, and 36 *Saccharum* spp. hybrids. A total of 624 polymorphic SSR alleles were amplified by PCR with 22 pairs of fluorescence-labeled highly polymorphic SSR primers and identified on a capillary electrophoresis (CE) detection system including 109 new alleles. Three approaches (model-based clustering, principal component analysis, and phylogenetic analysis) were conducted for population structure and genetic diversity analyses. The results showed that the 196 accessions could be grouped into either three (Q) or eight (q) sub-populations. Phylogenetic analysis indicated that most accessions from each species merged. The species *S. barberi* and *S. sinense* formed one group. The species *S. robustum*, *S. barberi*, *S.* spontaneum, *S. edule*, and sugarcane hybrids merged into the second group. The *S. officinarum* accessions formed the third group located between the other two groups. Two-way chi-square tests derived a total of 24 species-specific or species-associated SSR alleles, including four alleles each for *S. officinarum*, *S. spontaneum*, *S. barberi*, and *S. sinense*, five alleles for *S. robustum*. and three alleles for *Saccharum* spp. hybrids. These species-specific or species-associated SSR alleles will have a wide application value in sugarcane breeding and species identification. The overall results provide useful information for future genetic study of the *Saccharum* genus and efficient utilization of sugarcane germplasm resources in sugarcane breeding.

**Keywords:** sugarcane; *Saccharum*; population structure; SSR; genetic diversity

## 1. Introduction

Sugarcane is one of the largest commercial sugar crops [1]. It is not only a raw sugar production resource (80% sucrose), but also an important green-fuel resource (40% ethanol) in world agriculture, providing a vital role of economic growth and food security for the tropical and subtropical regions of the world [2]. Taxonomically, sugarcane is placed under the grass family *Poaceae*, subfamily *Panicoideae*, tribe *Andropogoneae*, subtribe *Sacharinae*, and genus *Saccharum* and shares genetic intimacy with *Sorghum* and other grasses. Sugarcane has gone through an extensive and complex process of domestication and hybridization [3]. The *Saccharum* genus contains six main species: the two wild species are *S. spontaneum* (2n = 40–128, x = 8) and *S. robustum* (2n = 60–80), and the four cultivated species are *S.*

*officinarum* (2*n* = 80, *x* = 10), *S. sinense* (2*n* = 111–120), *S. barberi* (2*n* = 81–124), and *S. edule* (2*n* = 60, 70, 80) [4].

Modern sugarcane varieties (*Saccharum* spp. hybrids) are widely complex polyaneuploid interspecific hybrids of these wild and cultivated species, with 110–130 chromosomes divided into ten homologous groups [5]. The continuing breeding efforts have led to a significant gain in cane yield and stress resistance, but a lesser gain on sucrose content due to the limited levels of genetic variations in sugarcane [6]. The limited introgression in sugarcane has resulted in a narrow genetic base of the current commercial varieties. Therefore, in sugarcane breeding, it is still a very important task to broaden the genetic base of sugarcane crops and improve stress resistance and sucrose content by using the gene pool of wild relatives.

Traditional diversity research at the morphological level is not only labor-consuming but also requires a high level of skill and breeding experience. Genetic diversity research at the molecular level has natural advantages over traditional morphological markers. Since the late 1980s, the molecular markers including amplified fragment length polymorphisms (AFLP) [7–9], restriction fragment length polymorphisms (RFLP) [10], random amplification of polymorphic DNAs (RAPD) [11], single nucleotide polymorphism (SNP) [12], simple sequence repeats (SSRs) [13,14], inter simple sequence repeat (ISSRs) [15], and expressed sequence tag-simple sequence repeat (EST-SSRs) [16–18] were being developed by breeders and geneticists and applied in many sugarcane studies. Due to its co-dominant, multi-allelic characteristics, relative abundance, and high genome coverage, SSR primers are one of the most effective markers in plant genetics and breeding [19,20] and have been widely used to study sugarcane genetic diversity, genetic mapping, cross-transferability, paternity analysis, segregation analysis, and marker-assisted selection [21].

Based on analysis of agronomic traits and mitochondrial profiles, *S. barberi* and *S. sinense* were placed in adjacent clusters, but apart from *S. robustum* [22]. A few genetic reports suggest that *S. robustum* be the progenitor of both *S. officinarum* and *S. edule*. Most sugarcane cultivars, along with *S. sinense* and *S. barberi*, are interspecific hybrids between *S. officinarum* and *S. spontaneum* [23]. Around 32–39% of the *S. sinense* and *S. barberi* genomes come from *S. spontaneum*, while for modern cultivars, the percentage is around 10~20% [10,24]. However, the interspecific relationship is not completely clear and there is no report on species-specific marker development in sugarcane for species identification.

A few reliable methods of population structure analysis were also developed including model-based clustering [25], principal component analysis, kinship analysis [26], and phylogenetic analysis [27], while genome instability is another matter for genetic analysis. de Araujo et al. (2005) reported that sugarcane had a higher level of the transcriptome (2.3%) compared with the 0.014% of maize and was undergoing the hybridization process between two polyploid species [28]. Due to high polyploidy with a high degree of chromosome instability, the sugarcane has given rise to a high degree of admixture and heterogeneity in some populations, which makes it hard to adequately define the structural complexity [17].

A better understanding of the population structure of different parental resources is the basis of genome-wide association study (GWAS) and helps tap into the wild relatives to broaden the genetic base [21,29,30]. This study aimed to characterize the population structure of the *Saccharum* genus through different approaches, including model-based clustering, principal component analysis, and phylogenetic analysis. The results may provide invaluable information to the future genetic study of the *Saccharum* genus and efficient utilization of sugarcane germplasm resources in sugarcane breeding.

## 2. Materials and Methods

### 2.1. Plant Materials

One hundred and ninety-six sugarcane accessions were used in this study, including 34 accessions from *S. officinarum*, 69 from *S. spontaneum*, 17 from *S. robustum*, 25 from *S. barberi*, 13 from *S. sinense*, 36 from *Saccharum* spp. hybrids, and two from *S. edule*. The leaf samples of all accessions were collected from either the Sugarcane Germplasm Nursery in

Yancheng, Hainan, China, or the World Collection of Sugarcane and Related Species at the USDA-ARS, Subtropical Horticulture Research Station, Miami, FL, USA (Supplementary Table S1). The leaf samples were placed on ice during collection and were kept in a −20 °C freezer until DNA extraction [31].

## 2.2. DNA Extraction

Genomic DNA was extracted from leaf tissues using a modified cetyltrimethylammonium bromide (CTAB) method as previously described by Pan (2006) [31]. The quality and concentration of DNA were measured using UV absorbance assay with either NanoDrop 1000 (Thermo Fisher Scientific, Waltham, MA, USA) or a Synergy™ H1 Multi-Mode Reader (BioTek, Winooski, VT, USA). The quality of DNA was further checked by 0.8% agarose gel electrophoresis [32].

## 2.3. SSR Markers and PCR Reactions

Twenty-two pairs of polymorphic SSR primers [31,33] were used in this study. All forward primers were labeled with fluorescence dyes, 6-carboxy-fluorescein (6-FAM), NED, or VIC. PCR reactions were conducted according to Chen [33]. The DNA fragments were identified by the capillary electrophoresis (CE) ABI 3730XL DNA Analyzer (Applied Biosystems Inc., Foster City, CA, USA) to generate GeneScan files following the manufacturer's instructions [34,35].

## 2.4. Marker Scoring

The GeneScan files were processed with the GeneMarker™ software (v. 2.80) (SoftGenetics LLC, State College, PA, USA, www.softgenetics.com, accessed on 27 December 2021) to reveal capillary electropherograms of SSR-DNA fragments with sizes calibrated against the GS500 DNA size standards (Applied Biosystems, Inc., Foster City, CA, USA). SSR alleles were manually assigned to unique, true "Plus-adenine" DNA fingerprints that gave quantifiable fluorescence values. Irregular peaks and stutters peaks were not scored according to Pan (2006) [31]. Data were scored manually in a binary format into a data matrix file, with the presence of a band scored as "1" and its absence scored as "0" and a binary format data were recorded manually as the presence of a band scored as "1" or "A" and its absence scored as "0" or "C" [36].

## 2.5. Structure and Genetic Diversity Analysis

STRUCTURE 2.3.1 [37] was used to infer the population structure. To identify the number of populations (K), the capturing of the major structure in the data, we set up at a burn-in period of 10,000 Markov Chain Monte Carlo iterations and 100,000 run length, with an admixture model following Hardy-Weinberg equilibrium and correlated allele frequencies as well as independent loci for each run. Ten independent runs were performed for each simulated value of K, ranging from 1 to 11. For each simulated K, the statistical value ΔK was calculated using the formula described by Evanno et al. (2005) [38]. The optimal K was determined using Structure Harvester [39].

Phylogenetic relationships and principal component analysis (PCA) were generated by TASSEL 5.2.13 to analyze genetic relationships among accessions and to determine the optimal number of clusters in the study. The number of principal components (PC) was chosen according to the optimum subpopulation determined in STRUCTURE 2, and a PCA plot was drawn using R package ggplot2 by the data from TASSEL 5 [40]. Genetic diversity also was assessed, and phylogenetic trees were drawn using MEGA 7 [41] based on the Maximum Likelihood tree method with the parameters as Shi et al. (2017) [42]. During the drawing of the phylogeny trees, the population structure and the cluster information were imported for the combined analysis of genetic diversity. For the sub-tree of each Q/q population, the shape of 'Node/Subtree Marker' and the 'Branch Line' were drawn with the same color as in the figure of the bar plot of the population clusters from the STRUCTURE analysis.

The phylogenetic tree is based on the genetic distance among the six species, *S. officinarum*, *S. spontaneum*, *S. robustum*, *S. barberi*, *S. sinense*, and *S. edule*, the cultivated sugarcane (*Saccharum* spp. hybrids) were calculated by using the neighbor-joining method and were visualized by software MEGA 7. Compared with other methods, the neighbor-joining method has a higher complexity and reliability in calculating genetic distance values. In addition, the number of PC was chosen according to the optimum subpopulation determined in STRUCTURE combined structured populations (Q-cluster), and a PCA plot was drawn using R package ggplot2 [43].

### 2.6. Species-Specific and Species-Associated SSR Allele

Four methods, single-marker regression (SMR), *t*-test, and the chi-square test or the percentage good-to-fit testing were used to identify species-specific and species-associated SSR alleles for *S. officinarum* (so), *S. spontaneum* (ssp), *S. robustum* (sr), *S. barberi* (sb), *S. sinense* (ssi), and *Saccharum* spp. hybrids (Hyb). *S. edule* (se) was not included because it had only two accessions included in this study. When we identified a species-specific or species-associated SSR allele, we gave a score of '9' for the accessions of the species and a score of '1' for all accessions in other species. In that way, we had six data sets for sugarcane hybrids and five *Saccharum* species. The marker data were recorded "A" as present and "G" as absent corresponding to "1" and "0". SMR was conducted using TASSEL 5. The *t*-test was performed for every single marker using visual basic codes in Microsoft Excel 2016.

For the chi-square test or the percentage good-to-fit testing, four combinations, '9.A', '9.G', '1.A' and '1.G' were divided for each allele in each group of the specific species based on our defined. The number of sugarcane accession for each combination in each species-group were recorded as n9(A), n9(G), n1(A), and n1(G), where n9(A) + n9(G) = n9, number of accessions in the target species, and n1(A) + n1(G) = n1, number of accessions in all other species except the target one. Four combination percentages were calculated as %n9(A) = 100 × n9(A)/n9; %n9(G) = 100 × n9(G)/n9; %n1(A) = 100 × n1(A)/n1; and %n1(G) = 100 × n1(G)/n1. Theoretically, a species-specific allele would be %n9(A) = 100; n9(G) = 0; %n1(A) = 0; and %n1(G) = 100. The two formulas for the chi-square test were used in this study. In the beginning, we used the below Formula (1), where 0.5 was used instead of 0.

$$\chi^2 = [n9(A) - n9] \times [n9(A) - n9]/n9 + n9(G) \times n9(G)/0.5 + n1(A) \times n1(A)/0.5 + [n1(G) - n1] \times [n1(G) - n1]/n1 \quad (1)$$

when we used this formula, we did not find any significant allele with $p \geq 0.01$. Late, we only test n9(A) and n1(G) in chi-square testing with below Formula (2) because the n9(G) and n1(A) were expected to be 0.

$$\chi^2 = [n9(A) - n9] \times [n9(A) - n9]/n9 + [n1(G) - n1] \times [n1(G) - n1]/n1 \quad (2)$$

Meanwhile, when the chi-square test cannot detect any species-specific allele at a significant level, we identified some associated alleles for the target species. If an allele had a >70% value in both %n9(A) and %n1(G) values, it was recognized as a species-associated allele. At the same time, the SMR and *t*-test can identify some species-associated alleles.

### 2.7. Phylogenetic Analysis of the Sugarcane Accessions Using Species Associated SSR Alleles

In this study, three top associated markers for each of the six species, plus six alleles that had 100% value either %n9(A) or %n1(G) were selected and used for phylogenetic analysis in the 196 accessions (Supplementary Table S2). Phylogenetic trees for the 196 accessions were drawn using MEGA 7 [44] based on the Maximum Likelihood tree method using seven allele sets, namely, (1) all species-associated alleles (1), *Saccharum* spp. hybrids (2), *S. barberi* (3), *S. officinarum* (4), *S. robustum* (5), *S. sinense* (6), and *S. spontaneum* (7), respectively.

## 3. Results

### 3.1. SSR Alleles

A total of 624 molecular alleles were amplified by PCR from the DNA of 196 accessions, 40.7%, 61.9%, 15.2%, 58.2%, 71.6%, 51.2%, and 77.6% of which were found in *Saccharum* spp. hybrids, *S. barberi*, *S. edule*, *S. officinarum*, *S. robustum*, *S. sinense*, and *S. spontaneum*, respectively (Supplementary Figure S1). Up to 46 alleles were amplified by each SSR primer pair. All alleles were highly polymorphic. Overall, 109 alleles have never been reported before, of which 26 alleles in sizes from 79 to 189 bp were amplified from a new SSR primer pair SMC319CG (Table 1).

**Table 1.** The general utility and amplification profile of 22 pairs of fluorescence-labeled SSR primers based on a capillary electrophoresis (CE) detection platform.

| No. | SSR Markers | Size Range (bp) | Number of Detected Peaks (Alleles) | Number of Original Alleles | Number of New Alleles | New SSR Alleles (bp) Detected in This Study |
|---|---|---|---|---|---|---|
| 1 | SMC119CG | 80~239 | 32 | 6 | 26 | 80, 92, 95, 99, 109, 115, 121, 125, 136, 138, 141, 143, 145, 149, 152, 155, 168, 172, 187, 190, 195, 204, 210, 229, 231, 239 |
| 2 | SMC1604SA | 91~140 | 26 | 19 | 7 | 92, 97, 104, 107, 126, 129, 140 |
| 3 | SMC1751CL | 122~169 | 26 | 23 | 3 | 156, 168, 169 |
| 4 | SMC18SA | 127~165 | 14 | 13 | 1 | 128 |
| 5 | SMC22DUQ | 123~177 | 21 | 20 | 1 | 141 |
| 6 | SMC24DUQ | 119~144 | 17 | 15 | 2 | 119, 126 |
| 7 | SMC278CS | 110~238 | 32 | 30 | 2 | 112, 144 |
| 8 | SMC319CG | 79~189 | 26 | 0 | 26 | 79, 123, 143, 144, 145, 147, 149, 151, 153, 155, 158, 160, 162, 163, 164, 166, 168, 170, 172, 175, 179, 181, 183, 185, 187, 189 |
| 9 | SMC31CUQ | 108~234 | 30 | 29 | 1 | 151 |
| 10 | SMC334BS | 130~170 | 30 | 27 | 3 | 130, 141, 169 |
| 11 | SMC336BS | 112~239 | 31 | 30 | 1 | 152 |
| 12 | SMC36BUQ | 82~338 | 35 | 27 | 8 | 102, 124, 137, 147, 192, 258, 277, 338 |
| 13 | SMC486CG | 222~247 | 15 | 14 | 1 | 229 |
| 14 | SMC569CS | 165~238 | 38 | 36 | 2 | 177, 211 |
| 15 | SMC597CS | 128~188 | 46 | 45 | 1 | 177 |
| 16 | SMC703BS | 181~237 | 33 | 31 | 2 | 181, 184 |
| 17 | SMC7CUQ | 144~171 | 16 | 13 | 3 | 144, 147, 150 |
| 18 | SMC851MS | 127~158 | 31 | 31 | 0 | N/A |
| 19 | CIR3 | 148~467 | 34 | 26 | 8 | 166, 401, 424, 440, 442, 462, 464, 467 |
| 20 | CIR43 | 162~256 | 37 | 31 | 6 | 162, 164, 203, 222, 249, 251 |
| 21 | CIR66 | 102~154 | 27 | 27 | 0 | N/A |
| 22 | CIR74 | 118~245 | 27 | 22 | 5 | 118, 175, 200, 202, 245 |
| Total | N/A | N/A | 624 | 515 | 109 | 109 |

### 3.2. Population Structure

The 196 sugarcane accessions could be classified into three or eight main populations using STRUCTURE 2.3.4 when the peaks of ΔK were observed at K = 3 and 8 (Figure 1A). The model-based structure classification of accessions into populations is shown in Figure 1B and Supplementary Table S1. At K = 3 with a likelihood threshold of 60% for cluster division, 191 accessions (97.4%) were grouped into three populations. The first cluster of 79 (40.3%) accessions was grouped into Population Q1; 27 (13.8%) into Population Q2; and 85 (43.4%) into Population Q3 (Figure 1B, Table 1 and Table S1). The remaining 5 accessions (2.6%) were placed in the admixture (Supplementary Table S1). In Population Q1, there were 69 (87.3%) *S. spontaneum*, seven *S. barberi* (8.9%), and four *S. robustum* (5%) accessions; in Population Q2, there were 16 (59.3) *S. barberi*, 10 (37.0%) *S. sinense*, and one (3.7%) *S.*

*robustum* accessions; and in Population Q3, there were 35 (41.2%) *Saccharum* spp. hybrids, 33 (38.8%) *S. officinarum*, 11 (12.9%) *S. robustum*, two (2.3%) *S. barberi*, two (2.3%) *S. edule*, and one (1.2%) *S. sinense* accessions. Population Q3 was divided into two sub-categories: high Q1-likelihood (0.3~0.4) part and low Q1-likelihood (<0.1) part, based on the proportion of the Q1 component. It is worth noting that all *S. officinarum* accessions belonged to the high Q1-likelyhood part.

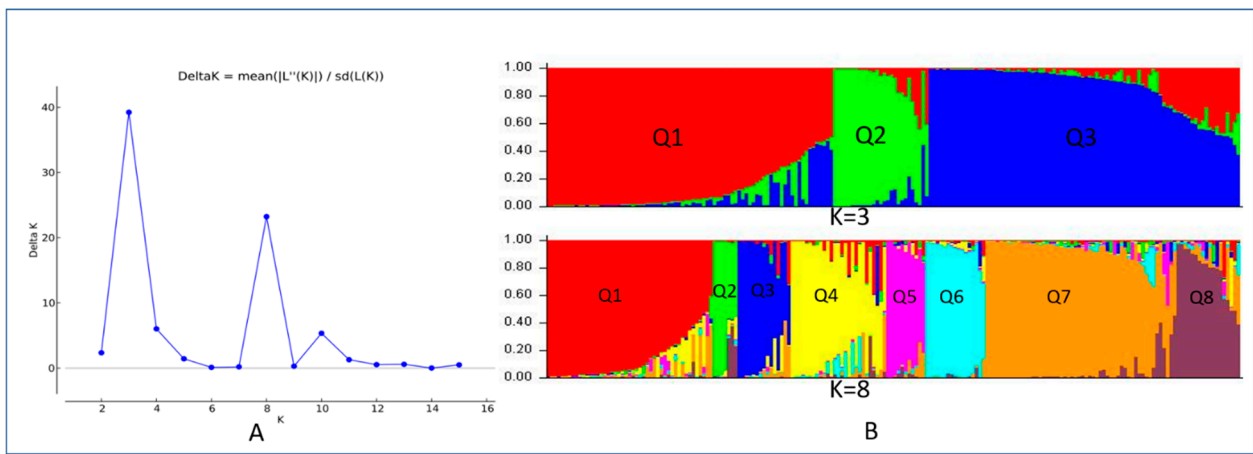

**Figure 1.** (**A**) Delta K values for different numbers of populations assumed (K) in the STRUCTURE analysis. (**B**) Classification of 196 accessions into three "Q populations" (K = 3) or eight "q populations" (K = 8) using STRUCTURE 2.3.1. The distribution of the accessions to different populations is indicated by the color code. Numbers on the *y*-axis show the subgroup membership, and the *x*-axis shows the different accession.

At K = 8 and a likelihood threshold value of 0.55, 188 (95.8%) accessions were grouped into eight populations (Figure 1B). The first population of 46 (23.4%) accessions was grouped into q1, the next seven (3.5%) into q2, 14 (7.1%) into q3 (Figure 1B and Supplementary Table S1), 24 (12.2%) into q4, 11 into q5 (5.6%), 17 into q6 (8.6%), 52 into q7 (26.6%), and 16 into q8 (8.1%). The remaining eight (4.0%) accessions were admixtures (Supplementary Table S1). In the q1 population (46 accessions), 35 (76.0%) were *S. spontaneum* accessions, nine (19.5%) were *S. officinarum* accessions, one (2.1%) was *S. barberi* accession, and one (2.1%) was *S. spontaneum* accession. In the q2 population, six were *S. barberi* accessions, and one was *S. sinense* accession. In the q3 population (14 accessions), 10 (71.4%) were *S. sinense* and four (18.6%) were *S. barberi*. In the q4 population (24 accessions), all accessions were *S. officinarum*, except for one (4.1%) *S. sinense* and one (4.1%) *S. robustum*. In the q5 population (11 accessions), all accessions were *S. barberi*. In the q6 population (17 accessions), all accessions were *S. robustum*, except for two *S. edule* (11.7%) and one (5.8%) *S. officinarum*. In the q7 population (52 accessions), all accessions were *S. spontaneum*. In the q8 population (16 accessions), all accessions were *S. spontaneum*.

*3.3. Phylogenetic Analysis*

Phylogenetic cluster analysis from MEGA 7 divided the 196 accessions into four groups, namely, Cluster I, II, III, and IV (Figure 2). Although there was no clear demarcation in the clustering pattern, this result was consistent with the model-based population structure (Q populations). We lined out all clusters by several curves which were marked by colors using as Q (1.2.3) at K = 3 in Figure 1. Cluster I (red curve) was further subdivided into three sub-clusters. Sub-cluster i included all *S. spontaneum* accessions and sub-clusters ii and iii contained 19 (76.0%) of *S. barberi* accessions. Except for sub-cluster-ii (green dotted curve), which was marked as Q2 (green), everything else in Cluster I matched with Q1 (red). Cluster II included all *S. officinarum* accessions, which were mentioned as a high Q1-likelihood part of the Q3 (blue) population. Almost all *S. sinense* accessions belonged to

Cluster III (green curve), which formed the Q2 (green) population with those accessions from sub-cluster ii (green dotted curve). Cluster IV (blue curve) was comprised of *S. robustum* and *S. officinarum* accessions, which were further divided into two sub-clusters (iv and v). These two sub-clusters were 97.4% matched with the low Q1-likelihood part of the Q3 (blue) population.

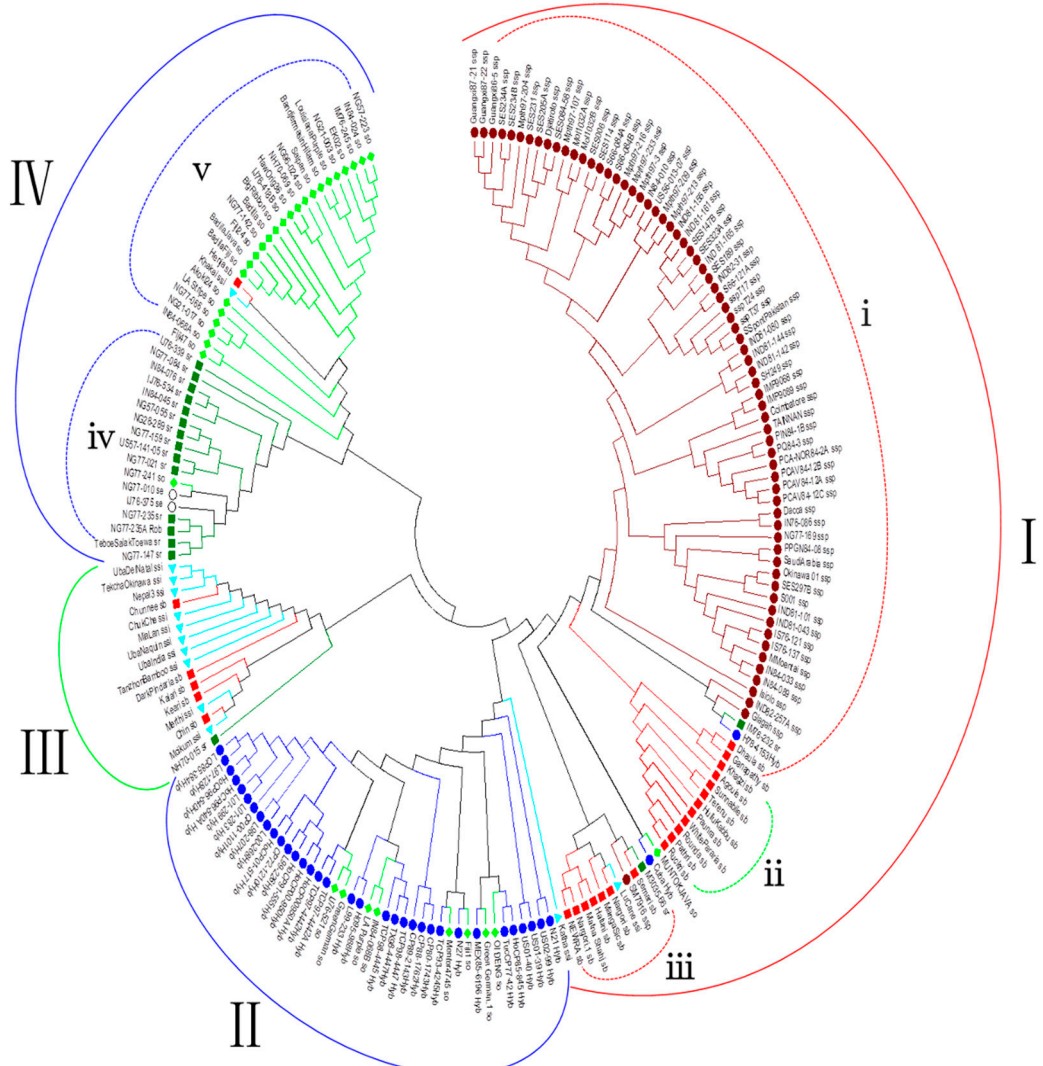

**Figure 2.** Phylogenetic tree for 196 accessions created by MEGA 7 based on SSR data. Each accession was marked by species as below: *S. spontaneum* (tan round), *S. robustum* (green square), *S. officinarum* (green diamond), *S. sinense* (blue triangle), *S. barberi* (red square), *Saccharum* spp. hybrids (blue round), and *S. edule* (white round). Four main clades are marked as **I**, **II**, **III**, and **IV** by full curves; five sub-clades are marked as **i**, **ii**, **iii**, **iv**, and **v** by dotted curves, with corresponding colors (red, green, and blue) of K3 gene pools (Q1, Q2, and Q3).

The result from phylogenetic analysis (Figure 3) was highly consistent with the phylogenetic analysis of accessions in Figure 1. The *S. barberi* and *S. sinense* accessions were placed in the same branch apart from other *Saccharum* species. The two wild *Saccharum* species, *S. spontaneum* and *S. robustum* were placed in the same sub-branch and away from the cultivated species *S. officinarum*. The sugarcane accessions were gathered in a branch in-between *S. officinarum* and the two wild species, due to their hybrid origin from these three species.

### 3.4. Principal Component Analysis

The distribution of all 196 accessions over the two PCA dimensions was also clustered based on K = 3 and K = 8 (Figure 4). Not only did these 196 accessions scatter in accordance with their taxonomic classification, but the PCA plot was consistent with the results of structure and phylogenetic analysis as well.

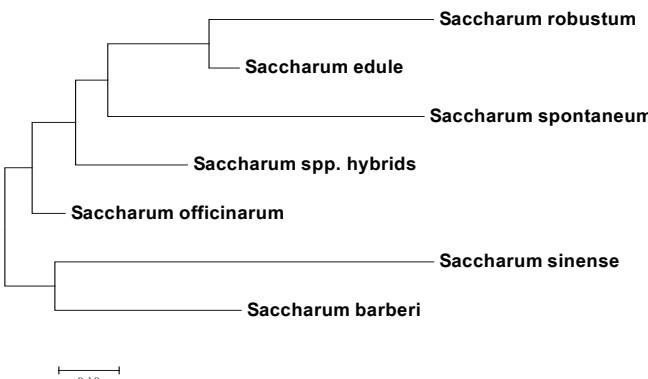

**Figure 3.** Phylogenetic tree among *Saccharum* spp. hybrids and six related *Saccharum* species: *S. spontaneum*, *S. robustum*, *S. officinarum*, *S. sinense*, *S. barberi*, and *S. edule* based on the genetic distance matrix using the neighbor-joining method by Power Marker v. 3.25 and visualized using the software MEGA 7.

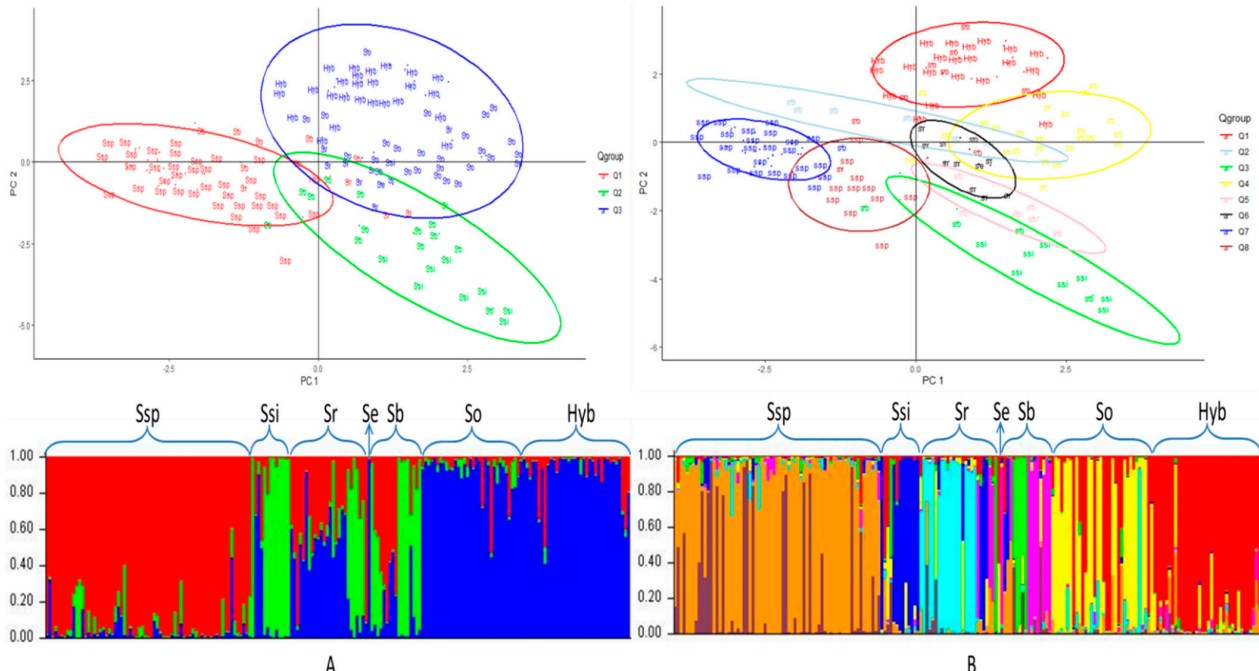

**Figure 4.** A scatter diagram of Principal Component Analysis (PCA) of the 196 accessions labeled by species as *Saccharum spontaneum* (ssp), *S. robustum* (sr), *S. officinarum* (so), *S. sinense* (ssi), *S. barberi* (sb), *Saccharum* spp. hybrids (Hyb), and *S. edule* (se), and marked by corresponding colors of K3 (**A**) and K8 (**B**) gene pools.

### 3.5. Species-Specific and Species-Associated SSR Alleles

The 4-way chi-square test by Formula (1) showed that no allele had a *p*-value of > 0.01, suggesting we did not identify any strong species-specific SSR alleles (Supplementary Table S2). However, when we used the 2-way chi-square test by Formula (2), eight alleles had >0.01 *p*-values, among which, 569CS_222 was specific to Hyb; 334BS_144 was specific to sb; 278CS_140, 31CUQ_136, 569CS_204, and 597CS_138 were specific to ssi; 703BS_223 and

703BS_225 were specific to sr (Supplementary Table S2). In addition, 16 alleles were identified to be species-associated based on a high percentage of %n9(A)/n9, or %n1(G)/n1, or both, and a high percentage of %(n9(A)/n9 + %n1(G)/n1)/2, or 100% in one of %n9(A)/n9 and %n1(G)/n1 (Supplementary Table S2). The 24 alleles have a high LOD ($-$log ($p$-value)) in SMR and $t$-test (Supplementary Table S2), indicating that the 24 alleles were species-specific or species-associated alleles.

### 3.6. Phylogenetic Analysis of the Sugarcane Accessions Using Species-Specific or Species-Associated SSR Alleles

Phylogenetic analyses in 196 sugarcane accessions were performed by seven sets of marker alleles: (1) all 24 selected alleles, (2) three Hyb-associated alleles, (3) four sb-associated alleles, (4) four so-associated alleles, (5) five sr-associated alleles, (6) four ssi-associated alleles, and (7) four ssp-associated alleles (Supplementary Table S2). Seven phylogenetic trees were drawn from each of the seven allele sets. The results showed that these 24 alleles could create a phylogenetic tree that grouped the sugarcane accessions by species (Figure 5B), which was similar to the phylogenetic tree with all 624 alleles (Figure 5A). According to Supplementary Figure S2, these species-specific or species-associated alleles can group accessions of a target species away from other species, although these makers cannot completely distinguish all accessions of certain species from those accessions in other species.

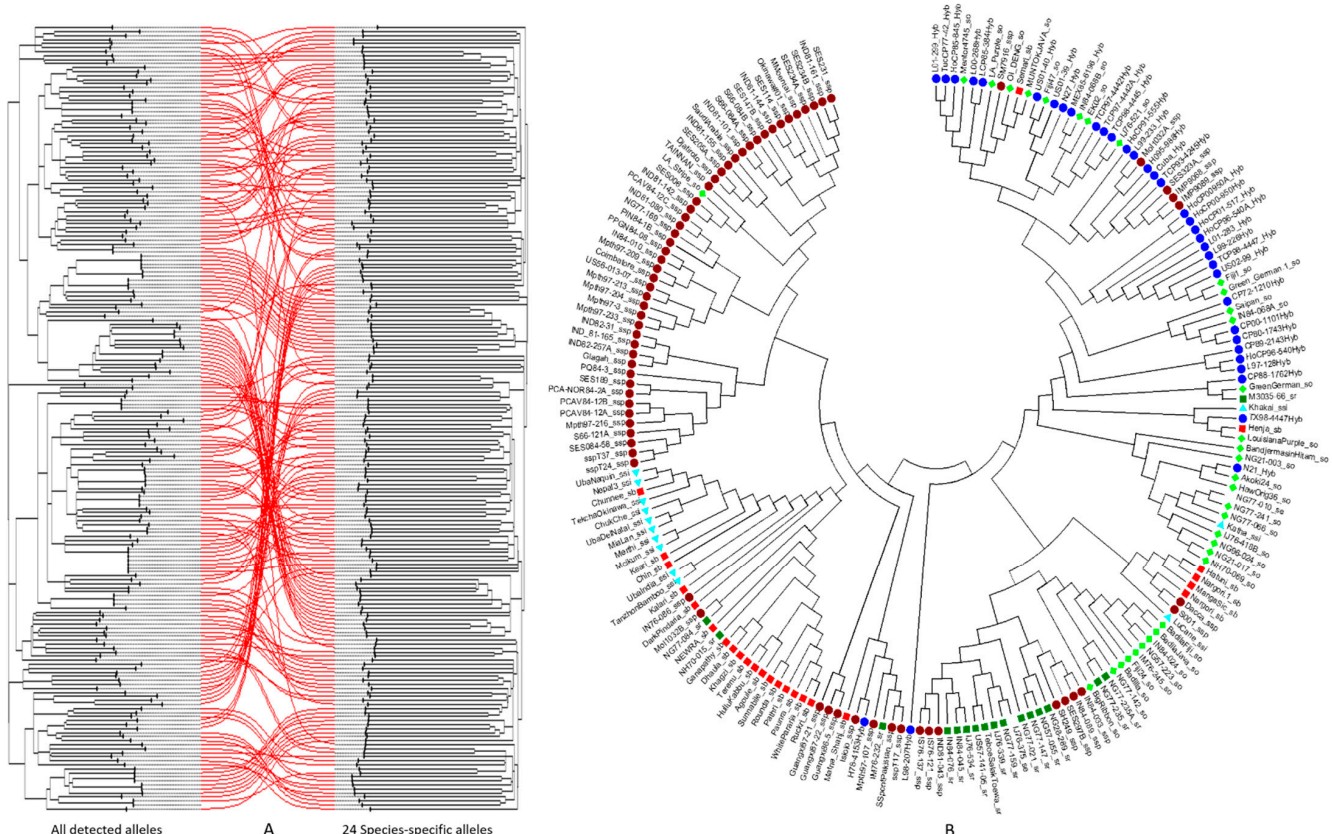

**Figure 5.** (**A**): Two phylogenetic trees of 196 accessions constructed using all 624 SSR alleles vs. 24 Species-specific alleles created from MEGA 7 based on 24 species-specific or species-associated alleles. (**B**): Circled phylogenetic tree of 196 accessions created by 24 species-specific or species-associated alleles with the labels as *S. spontaneum* (tan round), *S. robustum* (green square), *S. officinarum* (green diamond), *S. sinense* (blue triangle), *S. barberi* (red square), *S. edule* (white round), and *Saccharum* spp. hybrids (blue round).

## 4. Discussion

### 4.1. Profile of the SSR Alleles

The 196 accessions included in the present study represent six *Saccharum* species and *Saccharum* spp. hybrids with different characteristic agro-economic traits [30,45]. These accessions were selected from the sugarcane germplasm collections in Mainland China and USA. Some are the progenitors of sugarcane cultivars that have been exploited extensively in sugarcane breeding programs [46]. Compared with the previous report [3], the outcome of this study was greatly improved in analytical methods with 87 new accessions and 457 new alleles. Especially in the structure analysis, this study showed a higher level of consistency among the various analyses compared with all previous studies, which greatly improved the value of the developed markers and germplasm [47].

SSRs have increasingly become the marker of choice for such genetic analyses as linkage [18] and gene mapping [48], segregation [49], population structure [46,50,51], marker-assisted selection [51], genetic relationships [52], marker-assisted backcrosses [44], and population genetics and phylogeny [33]. Being highly polymorphic and more economic, SSRs are still preferred markers for species with complex genomes such as sugarcane [50]. 624 polymorphic SSR alleles were amplified with 22 pairs of SSR primers, averaging 28.3 alleles per primer pair. This level of polymorphism was higher than most genetic diversity studies reported earlier, for example, 205 alleles with 13.67 per locus [14], 209 alleles with 5.8 per locus [30], and 261 alleles with 7.35 per locus [53]. Moreover, according to the distribution difference of the SSRs between species, the markers "SMC1751CL", "CIR43", "SMC486CG", "SMC278CS", "SMC24DUQ", "SMC22DUQ" and "SMC18SA" were high identification and diversity SSRs (Supplementary Figure S1). Such efficient genetic markers can not only offer an accurate description of individual accession within a population or species, but more importantly, have great potential for further population structure analysis, and provide high-quality genetic marker information for breeding, GWAS, and evolution related studies in the future [21].

### 4.2. Population Structure and Phylogenetic Analysis

The population structure analyses divided the 196 accessions into 3 and 8 gene pools based on the peak of ΔK at K = 3 and 8. The ΔK showed a higher score at K = 2, 3, 5, or 7 in earlier studies using different sugarcane panels [21,46,50,54]. In Figure 2, most accessions in Q1 (K = 3) belong to two species, *S. spontaneum* or *S. barberi*, except for "MUNTOKJAVA" (*S. officinarum*) and "H78-4153" (*Saccharum* spp. hybrids). A similar phenomenon was observed not only in other "Q populations", but also in the "q populations" when K = 8. For example, 35 of 36 *Saccharum* spp. hybrids belonged to q1; 94.5% accessions of q2 and q5 were *S. barberi*; all *S. officinarum* accessions belonged to q4; all accessions in q7 and q8 are *S. spontaneum.* Although the 22 pairs of SSR primers are highly efficient, it is still necessary to argue that population structure analysis alone is based on probability and cannot provide the immaculate divisions among all clusters [17]. Therefore, neither K = 3 nor K = 8 can perfectly cluster or segregate all 196 accessions, and the forgoing conclusions must be combined with application scenarios in the future.

Based on the population structure analysis by STRUCTURE (Figure 1), the three gene pool systems were generally fit and delivered good results based on the species using genetic distance analysis by MEGA to create phylogenetic trees (Figures 2 and 3). However, these analyses could not distinguish each species. There were five sub-clades/clusters (Lowercase roman) in the phylogenetic tree under the four main clades/clusters (Uppercase roman). All these clades could cluster the accessions by species in this study almost perfectly and match the gene pool dividing by K = 8. It is worth noting that the four main clades were created based on the same genetic distance, while the five sub-clades were divided according to the standard of each main clade, not the unified standard of the whole phylogenetic tree. In other words, although the phylogenetic analysis showed a very clear pattern of species segregations, the immediate species-based division of phylogenetic trees still cannot be achieved by a unified genetic distance but must be accomplished by re-

division under the main clades. This fact also proved that the two clustering modes (K = 3 or K = 8) in structure analysis were not contradictory but complementary to each other [17]. The phylogenetic analysis for species relationship (Figure 4) is consistent with those of all accessions (Figure 3), and the differences may be caused by the number, diversity, and variation of the accessions within each species [33].

As shown in the PCA diagram, all 196 accessions displayed an obvious clustering tendency according to their species or Q (q) populations (K = 3 or K = 8), which was highly consistent with the phylogenetic trees. However, there also were a lot of mixture areas over all the clusters, which may help describe the relationship between species or Q (q) populations. PCA diagram is the most visualized clustering analysis, but due to the bias of the dimension reduction, the discrimination between each cluster was weakened [55], so was the precision of the generated 2D diagram [56]. The most typical case is that there would be many scatter overlaps or approximate overlaps in the graph, which result in the decay of the distance between each cluster area. For this reason, it is hard to see independent distribution patterns with clear boundaries. The same phenomenon also appears in this study, therefore when using these data we should also combine the results of other cluster analyses to draw a comprehensive conclusion [57].

### 4.3. Species-Specific and Species-Associated SSR Alleles

Due to the abundant level of polymorphism, SSR markers have received great attention in searching for species-specific alleles. In this study, 24 out of the 624 SSR alleles were found to be species-specific or species-associated and useful in identifying or differentiating different sugarcane species. Only three to five specific alleles were needed to identify most accessions of a particular species. This result will undoubtedly improve the efficiency of using SSR markers in identifying species in the future. However, it is still not possible to precisely describe the relationship between the accessions within or outside each species due to the very small genetic distances among them. Therefore, these markers must be used flexibly considering the application scenario.

### 4.4. Summary

Genome-wide association studies have been widely conducted to identify molecular markers associated with many valuable traits in many crops [58]. Due to the complex relationship and genetic background between genotypes in breeding populations, it is necessary to separate true marker-trait associations from the marker-trait association due to population structure [59]. The methods of population structure analysis include model-based clustering, principal component analysis, kinship analysis, and phylogenetic analysis. If one applies just one method to interpret population structure, the conclusion can be ineffective or biased due to the failure of capturing the entire complexity of the population, especially for a genetically complex polyploidy crop like sugarcane [60]. Therefore, model-based clustering, principal component analysis, and phylogenetic analysis were conducted in this study to comprehensively analyze and discuss the population structure of 196 sugarcane accessions based on 624 SSR alleles amplified by 22 pairs of SSR primers. The overall results will provide a valuable reference for sugarcane germplasm management and genetic improvement through breeding.

## 5. Conclusions

Three (Q) and eight (q) well-differentiated populations were postulated from this study among the 196 accessions of sugarcane and six related *Saccharum* species based on the binary data of SSR alleles. A total of 624 alleles were amplified by PCR with 22 pairs of highly polymorphic SSR primers, of which 109 alleles were new and 24 alleles were species-specific or species-associated alleles. Almost all accessions could be appropriately grouped to a specific cluster according to their species. The subsequent PCA and phylogenetic analyses also drew similar conclusions, which further verified the reliability of the results. Due to the importance of population structure analysis in diversity analysis, the

information generated in this research will help improve future sugarcane breeding programs towards good sugar recycling, paper and plywood, winemaking, and other related industrial areas. Moreover, the molecular data from this study will provide an important means for designing introgressive hybridization, making impactful cross combinations, and developing other related breeding strategies.

**Supplementary Materials:** The following supporting information can be downloaded at: https://www.mdpi.com/article/10.3390/agronomy12020412/s1, Figure S1: Distribution of the 624 SSR alleles amplified with 22 pairs of SSR primers from *Saccharum* spp. hybrids and six related *Saccharum* species. Figure S2: Phylogenetic trees for six *Saccharum* species based on species-specific or species-associated SSR alleles. Table S1: The information of 196 sugarcane accessions of *Saccharum* spp. hybrids and six related *Saccharum* species. Table S2: Chi-square test, the percentage good-to-fit testing, single-marker test, and *t*-test of 24 species-specific or species-associated SSR alleles among 196 *Saccharum* accessions.

**Author Contributions:** Y.-B.P. and S.-J.G. conceived the project, designed the experiments, conducted the SSR fingerprinting, and collected the original data. H.X., A.S. and Y.C. organized and analyzed the original data. H.X. drafted the manuscript. Y.-B.P., S.-J.G. and A.S. critically revised the manuscript. All authors have read and agreed to the published version of the manuscript.

**Funding:** This work was supported by the competitive grants of Sugarcane Germplasm Committee, the USDA-ARS, NPL; a USDA-ARS Non-Assistance Cooperative Agreement on Genetic Analysis and Trait-Specific Molecular Marker Development (Accession No. 440501); and the China Agriculture Research System of MOF and MARA (Grant No. CARS-17).

**Institutional Review Board Statement:** Not applicable.

**Informed Consent Statement:** Not applicable.

**Data Availability Statement:** Not applicable.

**Acknowledgments:** We thank Lionel Lomax and Sheron Simpson for their excellent technical support. The authors are thankful to Yulin Jia (USDA-ARS, Stuttgart, AR), Zhongqi He (USDA-ARS, New Orleans, LA, USA), and Perng-Kuang Chang (USDA-ARS, New Orleans, LA, USA) for reviewing the manuscript with excellent editorial comments. USDA is an equal opportunity provider and employer.

**Conflicts of Interest:** The authors declare no conflict of interest.

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
