# Peer review of "Population Structure and Genetic Diversity Analysis in Sugarcane (Saccharum spp. hybrids) and Six Related Saccharum Species"

_agronomy, doi:10.3390/agronomy12020412_

Round 1

Reviewer 1 Report

In general, an important contribution for sugarcane biotechnology. The manuscript has omissions and requires minor revisions.  You could avoid abbreviations after a point and include meaning of the missing initials.

- In INTRODUCTION section, the word “Gramineae” should be deleted and update references (2020-2021).

-In MATERIALS AND METHODS section, the authors could briefly explain why they used neighbor-joining method and not another genetic distance.

In RESULTS, you might consider removing the Table 2 Chi-square tests or include this in supplemental material.

In DISCUSSIONS section,

-The discussion should be ordered by topic and at the end add a general discussion.

Author Response

Point-to-Point Responses from the authors

  1. Review Report from Reviewer 1: 
  • In general, an important contribution for sugarcane biotechnology. The manuscript has omissions and requires minor revisions. You could avoid abbreviations after a point and include meaning of the missing initials.

Authors’ response: Thank you very much for your time and excellent comments. We appreciate your kind compliments and fully accommodated your suggestions.

  • In INTRODUCTION section, the word “Gramineae” should be deleted and update references (2020-2021).

Authors’ response: “Gramineae” was deleted and the references [3], [5], [7], [8], [10], [16] published in 2020~2022 were updated.

  • In MATERIALS AND METHODS section, the authors could briefly explain why they used neighbor-joining method and not another genetic distance.

Authors’ response: A brief explanation was added in lines 173-174.

  • In RESULTS, you might consider removing the TableChi-square tests or include this in supplemental material.

Authors’ response: Thank you so much! Table 2 was deleted from the manuscript and all citations of Table 2 were changed to Supplementary Table S2, which contains more detailed data.

  • In DISCUSSIONS section, the discussion should be ordered by topic and at the end add a general discussion.

Authors’ response: OK. Four subtitles were added to the Discussion including a summary.

Reviewer 2 Report

In this work, 22 pairs of highly polymorphic SSR primers by using on a capillary electrophoresis (CE) detection system including and a total of 624 polymorphic SSR alleles were amplified among 196 Saccharum accessions, including 34 S. officinarum, 69 S.spontaneum, 17 S.robustum, 25 S.barberi, 13 S.sinense, 2 S.edule, and 36 Saccharum spp. hybrids.

In addition, the authors claim that 109 new alleles were identified for the SSR primers used.

Typically, SSR markers are developed for a specific species and are effective for intraspecific analysis. SSR markers are not used for interspecies analysis. First, the sequence of microsatellites for other species will be different and not inherited between species. Second, polymorphic bands for one species will differ from another. Comparison of bands of different lengths for different species does not allow their use in phylogenetic analysis. Since phylogenetic analysis requires common bands as the main condition. But the SSR method is species-specific, and the bands between different species will be of different lengths and not the same size.

In addition, for very closely related species, the use of SSR markers is potentially possible, but with a preliminary analysis of the obtained bands, which need to be sequenced.

The authors showed that 22 pairs of highly polymorphic SSR primers identified up to 624 polymorphic SSR bands. Thus, there are, on average, 28 alleles for each SSR locus. These data indicate that the number of bands from different Saccharum species exceeds all possible limits, and thus show that phylogenetic analysis using these markers is impossible.

Authors should present a PAAG for PCR analysis of these SSR markers for reviewers and readers to appreciate the work.

Second, I have not seen the Supplementary Table S1 file, but I think that gel electrophoresis for the studied SSR markers will not be presented in this file.

Author Response

Point-to-Point Responses from the authors

  1. Review Report from Reviewer 2:
  • In this work, 22 pairs of highly polymorphic SSR primers by using on a capillary electrophoresis (CE) detection system including and a total of 624 polymorphic SSR alleles were amplified among 196 Saccharum accessions, including 34 S. officinarum, 69 S.spontaneum, 17 S.robustum, 25 S.barberi, 13 S.sinense, 2 S.edule, and 36 Saccharum spp. hybrids. In addition, the authors claim that 109 new alleles were identified for the SSR primers used.

Typically, SSR markers are developed for a specific species and are effective for intraspecific analysis. SSR markers are not used for interspecies analysis. First, the sequence of microsatellites for other species will be different and not inherited between species. Second, polymorphic bands for one species will differ from another. Comparison of bands of different lengths for different species does not allow their use in phylogenetic analysis. Since phylogenetic analysis requires common bands as the main condition. But the SSR method is species-specific, and the bands between different species will be of different lengths and not the same size.

Authors’ response: In general, SSR markers are ubiquitously distributed across the entire genome and are highly polymorphic and reliable. There are abundant reports on SSR marker-based genetic analysis including population structure and phylogeny, covering both intra- and inter-species, and in few cases, inter-genius plants. This paper deals with the Saccharum genus that has six commonly highly polyploidy species, including two wild species S. spontaneum (2n = 40–128, x = 8) and S. robustum (2n = 60–80), and four cultivated species S. officinarum (2n = 80, x = 10), S. sinense (2n = 111–120), S. barberi (2n = 81–124), and S. edule (2n = 60, 70, 80). Prior studies showed that SSR markers work well on all these species, even in related genus. One can find out these reports by searching the web using keywords “Saccharum”, “sugarcane”, “SSR”, etc.

  • Please see capillary electrophoregrams of six Saccharum species and sugarcane cultivars with two primer pairs SMC1604SA (green, left) and SMC597CS (blue, middle) (unpublished) and of 12 sugarcane cultivars with primer pair SMC336BS (black, right) (Pan, YB. 2016 Agronomy 2016, 6, 28; doi:10.3390/agronomy6020028).
  • In addition, for very closely related species, the use of SSR markers is potentially possible, but with a preliminary analysis of the obtained bands, which need to be sequenced.

Authors’ response: Sorry, but to our knowledge, there has not been any report on sequencing of SSR fragments in the literature. It is more costly to get the nucleotide sequence of SSR fragments in sugarcane because it requires cloning of these PCR-amplified SSR fragments first. Besides, sequencing is not the objective of this study. We can propose a research experiment to get some funding on this. 

  • The authors showed that 22 pairs of highly polymorphic SSR primers identified up to 624 polymorphic SSR bands. Thus, there are, on average, 28 alleles for each SSR locus. These data indicate that the number of bands from different Saccharum species exceeds all possible limits, and thus show that phylogenetic analysis using these markers is impossible.

Authors’ response: Yes, it is impossible for diploids or animal species. But, we are dealing with highly polyploids in Saccharum as explained earlier. This is the data we got . 

  • Authors should present a PAAG for PCR analysis of these SSR markers for reviewers and readers to appreciate the work.

Authors’ response: PAAG? What do you mean? If it is PAGE, then we can explain. We used fluorescence labeled SSR primer pairs to amplify SSR fragments from the DNA of sugarcane accessions and separated these SSR fragments through capillary electrophoresis. The CE system uses fluorescence-labeled DNA size standards in every sample to calibrate the sizes of the SSR fragments, which is not feasible for PAGE (Pan, YB, Scheffler, BS, and Richard, Jr., EP. 2007. Sugar Tech. 9: 176-181). Please also view the capillary electrophoregrams shown earlier.

  • Second, I have not seen the Supplementary Table S1 file, but I think that gel electrophoresis for the studied SSR markers will not be presented in this file.

Authors’ response: Sorry that you were not able to see the Supplementary Table S1 file. It was included in the submission, and we do not get similar comments from other peer reviewers. In addition, we noticed that you have indicated that Moderate English changes required for this manuscript. Can you show us where the language deficiencies are? The Review/editor tool of the Microsoft Word software gives this manuscript a 99% score. We hope you will help us improve the language further to reach the 100% score.

Round 2

Reviewer 2 Report

The authors' comments satisfied my questions.